# Control of Redox Homeostasis by Short-Chain Fatty Acids: Implications for the Prevention and Treatment of Breast Cancer

**DOI:** 10.3390/pathogens12030486

**Published:** 2023-03-19

**Authors:** Carmen González-Bosch, Patricia A. Zunszain, Giovanni E. Mann

**Affiliations:** 1Departamento de Bioquímicay Biología Molecular, Universitat de València. Dr Moliner 50, 46100 Valencia, Spain; 2Department of Psychological Medicine, Institute of Psychiatry, Psychology & Neuroscience, King’s College London, 150 Stamford Street, London SE1 9NH, UK; 3King’s British Heart Foundation Centre of Research Excellence, School of Cardiovascular and Metabolic Medicine & Sciences, Faculty of Life Sciences & Medicine, King’s College London, 150 Stamford Street, London SE1 9NH, UK

**Keywords:** breast cancer, short-chain fatty acids, butyrate, microbiota, redox signaling, Keap1-Nrf2, epigenetics, antioxidants

## Abstract

Breast cancer is the leading cause of death among women worldwide, and certain subtypes are highly aggressive and drug resistant. As oxidative stress is linked to the onset and progression of cancer, new alternative therapies, based on plant-derived compounds that activate signaling pathways involved in the maintenance of cellular redox homeostasis, have received increasing interest. Among the bioactive dietary compounds considered for cancer prevention and treatment are flavonoids, such as quercetin, carotenoids, such as lycopene, polyphenols, such as resveratrol and stilbenes, and isothiocyanates, such as sulforaphane. In healthy cells, these bioactive phytochemicals exhibit antioxidant, anti-apoptotic and anti-inflammatory properties through intracellular signaling pathways and epigenetic regulation. Short-chain fatty acids (SCFAs), produced by intestinal microbiota and obtained from the diet, also exhibit anti-inflammatory and anti-proliferative properties related to their redox signaling activity—and are thus key for cell homeostasis. There is evidence supporting an antioxidant role for SCFAs, mainly butyrate, as modulators of Nrf2-Keap1 signaling involving the inhibition of histone deacetylases (HDACs) and/or Nrf2 nuclear translocation. Incorporation of SCFAs in nutritional and pharmacological interventions changes the composition of the the intestinal microbiota, which has been shown to be relevant for cancer prevention and treatment. In this review, we focused on the antioxidant properties of SCFAs and their impact on cancer development and treatment, with special emphasis on breast cancer.

## 1. Introduction

Cancer is considered a multi-stage process due to the accumulation of genetic and/or epigenetic alterations. These changes include the activation of different oncogenes and the inactivation of tumor suppressor genes, leading to the malignant transformation of healthy cells [1]. A small percentage of human cancers are linked to genetic inheritance, but the majority of genetic alterations are induced by aging, infections, chemical exposure, ultraviolet light and other environmental factors. 

Research has consistently established the contributory role of oxidative stress in human cancer, associated with the promotion of mutagenic DNA damage by elevated reactive oxygen species (ROS) [2,3,4,5,6]. Thus, when cellular antioxidants and detoxification enzymes are not sufficient to cope with this challenge, new ways to increase the antioxidant potential of cells and reduce ROS production could be useful to reduce cancer risks.

Breast cancer (BC) is one of the most well-known malignant tumors among women, and its incidence worldwide has risen worryingly in recent decades [7]. Therefore, there is an increasing need to identify new prevention strategies in order to reduce its incidence. BC is a complex disease, classified based on distinct gene expression signatures and histological characteristics [8,9]. 

Treatment with dietary substances with anticancer properties constitutes an attractive alternative in cancer therapy and complementary medicine **[10]**. In particular, there is current evidence that implicates the Mediterranean diet as an effective diet in primary prevention of BC [11]. The protective effects of this have been primarily associated with the intake of foods with important antioxidant properties due to their high content of polyphenols, flavonoids, carotenoids and fiber, with increasing evidence suggesting that the gut microbiota of subjects consuming such a diet differ significantly from those following a Western diet model. Gut microbiota are in a dynamic balance with the host, playing a crucial role in the maintenance of human health [12]. Recent research assessed the anticancer activity of gut microbiota metabolites, including the short-chain fatty acid (SCFA) butyrate, specifically against BC [13]. 

We recently revisited the role of specific SCFAs, produced by microbial fermentation or provided in the diet, in the regulation of host redox homeostasis, in order to highlight the link between microbiota, redox signaling, and host metabolism [14]. Herein, we reviewed, in detail, the in vitro and in vivo evidence supporting the anticancer and genotoxic effects of SCFAs in BC. We also discussed the demonstrated modes of action of these bioactive molecules, with a special emphasis on the regulation of redox homeostasis by specific SCFAs. We then critically evaluated the potential of these molecules for nutritional and therapeutic treatment of BC, alone or in combination with other forms of therapy, and their influence on the efficacy of standard chemotherapeutics. 

## 2. Natural Antioxidants in the Prevention and Treatment of Breast Cancer

Modern medicine emphasises the prevention and treatment of cancer with natural nutritional components. Among these, compounds with antioxidant properties have been introduced to target tumor cells with different modes of action [15]. Although the effect of these bioactive molecules on BC in in vitro studies is promising (see Table 1), in vivo studies and clinical trials remain inconclusive, highlighting the fact that the beneficial effects could depend on dose and treatment duration [16]. Notably, clinical trials, including dietary interventions rich in antioxidant nutrients, have shown a reduction in the prevalence and severity of fatigue, a side effect of BC, associated with chronic inflammation [17]. Antioxidants can act as extracellular ROS scavengers and/or modulating intracellular signaling pathways. In particular, nuclear factor erythroid 2 (NF-E2)-related factor 2 (NRF2) is a master regulator of redox homeostasis, controlling more than 200 genes [18,19]. NRF2 maintains homeostatic control of redox status in healthy cells, but can have a pro-carcinogenic effect in the case of transformed malignant cells [20].

Therefore, appropriately timed and targeted manipulation of the NRF2 pathway is critical when considering the development of effective therapeutics [21]—including plant-based NRF2-inducers, which help to protect humans from carcinogenic damage and NRF2 inhibitors, which help to counteract NFR2 overactivation in cancer—that impact clinical treatments [22]. In this section, we have presented information available for specific antioxidants.

Carotenoids are organic pigments that, together with their derivatives, constitute a group of molecules naturally produced by plants and other photosynthetic organisms. Their antioxidant properties have been studied due to their potential in preventing and treating multiple diseases, including cancer [23]. Among these, lycopene, found in red vegetables and fruits (especially in tomatoes), exhibits anti-tumor effects via regulation of cell growth factor signaling pathways, initiation of cell cycle arrest and induction of cell apoptosis [24]. In vitro studies with estrogen receptor (ER)-positive breast cancer MCF-7 cells showed that lycopene treatment reduced cell proliferation and increased apoptosis, possibly upregulating the expression of the tumor suppressor p53 and that of Bax, a pro-apoptotic member of the Bcl-2 family [25]. 

**Table 1 pathogens-12-00486-t001:** Beneficial effects of plant-derived antioxidants on breast cancer in vitro.

Compound	Cell Line	Treatment	Effects	Mechanism	Ref
Quercetin	MCF-7	100 µM	↓cell proliferation	↑ROS dependent apoptosis	[26]
	MC-F MDA-MB-468	75 μM (+100 μM VC)	↓cell proliferation	↑ROS ↓NRF2	[27]
Astaxanthin	MCF7	50 µM (24 h)	↓cell proliferation ↓cell migration	↑Bax ↑BCL-2	[28]
Lycopene	MCF7	2–16 µM (24–72 h)	↓cell proliferation ↑Apoptosis	↑p53 ↑Bax	[25]
	MDA-MB-468	100 µM (24–72 h)	↓cell proliferation ↑Apoptosis	↑Bax ↑ERK1⁄2 ↓D1 ↑p21	[29]
Resveratrol	MCF7	25–50 µM (72 h)	↓cell proliferation ↑Apoptosis	↓PI3K and Akt ↑caspase 3, 9	[30]
	MCF-7	100 µM (24 h)	↓cell proliferation	↑ROS ↓CK2	[31]
Sulforaphane	ZR-75-1	25 µM (72 h)	↓cell proliferation	G1/S arrest ↓SERTAD1, CCND2	[32]
	MCF7	25 µM (72 h)	↓cell proliferation ↑Apoptosis	G2/M arrest ↓HDACs ↑caspase 3, 9 ↓ER-α, EGFR, HER-2	[33]
Gingerol	MDA-MB-231	10 µM	Inhibition of metastasis	↓MMP-2 and MMP-9	[34]

Abbreviations: Akt: protein kinase B; Bax: Bcl-2-associated X-protein; BCL-2: B-cell lymphoma 2 protein; CCND2: cyclin D2; ER-α: estrogen receptor alpha; EGFR: epidermal growth factor receptor; HER-2: epidermal growth factor receptor-2; HDACs: histone deacetylase 2; MMP-2: matrix metalloproteinase-2; MMP-9: matrix metalloproteinase-9; p53: tumor suppressor p53; PI3K: phosphatidylinositol 3-kinase; ROS: reactive oxygen species; SERTAD1: SERTA domain-containing protein-1; VC: Vtamin C. Upward arrow (↑) indicates an increase, and downward arrow (↓) a decrease, in respective measured outcomes.

Lycopene, extracted from guava, also has cytotoxic effects on this breast cancer cell line, altering the cell cycle and increasing apoptotic cells, compared to untreated cells [35]. Of note, the C40 carotenoid astaxanthin has the highest oxygen radical scavenging capacity, and several studies have demonstrated antitumoral properties, supporting its therapeutic potential for the prevention or co-treatment of cancer [36]. It has been reported that carotenoid oxidation products are the active mediators in the stimulation of phase II detoxifying enzymes mediated by NRF2 signaling [37]. There is preclinical evidence of the preventive effects of lycopene on mouse cutaneous tumors in the promotion phase, via stimulation of NRF2 signaling pathway, and p62-mediated degradation of Keap1, via the autophagy-lysosomal pathway [38]. Recent studies have shown that the halophilic carotenoid bacterioruberin from haloarchae may have a promising future due to its particular structure and antioxidant activity [39].

Polyphenols in foods and dietary supplements are implicated in the prevention and treatment of BC [40]. Phytoestrogens are plant-derived polyphenolic compounds (including flavonoids and stilbenes, among others) with biological properties similar to those of human estrogens [41,42,43]. Due to structural resemblances, they can bind to estrogen receptors, providing either an estrogenic or an antiestrogenic action. Non-receptor-mediated effects include antioxidant effects and inhibition of enzymes involved in estrogen synthesis [44]. However, there is a paucity of information on their mode of action. Estrogens play key roles in breast cancer progression and the interaction between endogenous steroid hormones and natural dietary polyphenols is highly dependent on dose and time of therapy to avoid undesirable effects [45]. 

Quercetin, a flavonoid ubiquitously present in fruits and vegetables, constitutes one of the most common dietary flavonoids in the Western diet. Due to its health benefits, it is considered a useful natural compound in cancer prevention and therapy [46]. The anticancer properties of quercetin in breast cancer have been recently revisited, with reported effects on proliferation, angiogenesis, and apoptosis [47]. Of note, the antioxidant or pro-oxidant activites of quercetin depend on its concentration and on the redox state of cells [48,49]. In vivo, breast cancer models showed that quercetin inhibited angiogenesis, by suppressing calcineurinregulated pathway activation [26]. A synergistic role of quercetin and vitamin C in NRF2-dependent oxidative stress production in different breast cancer cell lines has been reported, noting dose-dependent actions [27]. In this study, sequential treatment with vitamin C and quercetin induced oxidative stress by reducing NRF2 mRNA, protein levels and nuclear translocation. These results suggested that polyphenols might also act as an adjuvant in combinatorial treatments for patients with cancer and overexpression of NRF2, because persistent NRF2-mediated antioxidant responses promote malignant progression, apoptotic resistance, and chemoresistance in cancer cells [50]. 

Aside from their antioxidant properties against the DNA-damaging effects of ROS, stilbenes also modulate cell-signaling pathways, altering the underlying factors that influence BC risk [40]. Resveratrol and other dietary polyphenols are inhibitors of estrogen metabolism in human BC cells, reducing cell proliferation [51]. Resveratrol is a plant-derived stilbene phytoestrogen present in various foods and beverages and showing antioxidant, detoxification, anti-inflammatory, and anticancer activities [52]. In vivo studies showed that resveratrol inhibited estrogen-induced breast carcinogenesis in rats via induction of NRF2-mediated protective pathways [53]. Recently, it was shown that resveratrol affected cell viability and mitochondrial function of MCF-7 cells via its pro-oxidant action, which inhibited casein kinase 2 (CK2) activity, critical in the proliferation and apoptosis of cancer cells [31]. Future mechanistic studies will be required to assess the therapeutic potential of polyphenols in inhibiting breast cancer initiation, promotion, and/or progression by acting as oxidative stress modulators via activation of NRF2 [54].

Sulforaphane (SFN) is an isothiocyanate obtained from cruciferous plants, such as broccoli. It induces antioxidant activities, inhibits cell proliferation, causes apoptosis, and stops the cell cycle by affecting histone deacetylases, gene expression, and NRF2 antioxidant signaling [55]. SFN is one of the most potent NRF2 inducers [56,57], contributing to carcinogen detoxification and reduction of oxidative stress [58] and reducing the growth of human breast cancer cells [59]. In vitro studies with SFN-treated BC cells have shown that it inhibits cell growth and activates apoptosis by inhibiting histone deacetylases (HDACs)[33]. The nutrigenomic potential of SFN in cancer therapy (by activating NRF2 signaling) has been demonstrated in combinatorial treatments with the anticancer acetazolamide. The latter has been associated with inhibition of the PI3K/Akt/mTOR survival pathway and induction of apoptosis in bronchial carcinoid cancer [60]. 

The chemoprotective effects of SFN in MCF-7 cells have been associated with reversing estrogen-induced metabolic changes and epigenetic regulation [61]. Recent studies showed that SFN regulated the miR-19/PTEN axis in MCF-7 cells to exert protective effects against BC promotion [62]. 

Broccoli sprouts are rich in the SFN precursor glucoraphanin, which is hydrolyzed by the plant enzyme myrosinase and by a similar enzyme present in the gut microbiota [63]. Therefore, SFN levels are highly dependent on dietary habits and microbiome composition. Although in vitro and in vivo studies have shown that SFN is effective in treating different stages of BC, further studies will be required to determine the precise dose and timing in individual and combinatorial treatments.

Accordingly, plant-derived bioactive agents are particularly interesting for researchers and clinicians due to their multiple modes of action in preventing and controlling BC, as well as their availability in our diets. Clinical trials suggest that such products could potential significantly increase the effectiveness of conventional antitumor agents—and decrease their side effects [64].

## 3. SCFAs Mediate Anticancer and Genotoxic Effects in Breast Cancer

We recently critically revisited the protection afforded by SCFAs against oxidative and mitochondrial stress, as well as their role in health and disease [14]. There is convincing evidence that SCFAs modulate redox homeostasis mainly via Keap1-NRF2 signaling, regulating NRF2 nuclear accumulation [65,66], increasing H3K9 acetylation via specific receptors [67], or through multiple molecular mechanisms with synergistic antioxidant effects [68]. These natural compounds, especially butyrate, exhibit anti-inflammatory and antiproliferative properties. These are associated with their redox signaling activity and depend on specific receptors and transporters [69]. In this section, we focused on selected experimental data supporting the role played by the redox signaling properties of butyrate and other SCFAs in inhibiting tumor progression through different mechanisms, with a special emphasis on BC.

### 3.1. Antiproliferative Properties of SCFAs

Changes in the gut microbiota in patients suffering colorectal cancer support a role for host–microbe interactions in the origin and development of malignancy. This has enabled the development of novel microbiota-based therapeutics and diagnostic tools [70,71]. Intake of dietary fiber, which has been associated with increased production of SCFAs (especially butyrate), inhibits human colon cancer cell proliferation via cell cycle arrest and apoptosis and may, therefore, protect against colon cancer [72,73]. SCFAs are crucial in cell homeostasis; therefore, the manipulation of SCFA levels in the intestinal tract via changes in microbiota composition could be relevant for cancer treatment/prevention. Recently, studies reviewed the detrimental and/or protective roles of SFCAs, produced by microbiota, in several cancers [74]. Butyrate seems to exert anticancer effects by modulating the immune system and through inhibition of multiple signaling pathways, including HDACs (see Table 2), although the mechanisms by which it mediates apoptosis and growth arrest in cancer cells remain unclear [75,76,77]. 

The activation of FFAR2 and FFAR3 receptors by SCFAs also drives BC cells toward a non-invasive phenotype and, therefore, may inhibit metastasis [82]. Many cancer cells exhibit altered acetylation levels, as well as overexpression of numerous HDACs [83,84]. The abnormal histone acetylation profile leads to cellular disorders, including tumor initiation and progression of BC [85]. In fact, to date, butyrate appears to be the most potent HDAC and tumor inhibitor among investigated natural compounds that epigenetically upregulate tumor-suppressor genes in cancer cells and anti-inflammatory genes in immune cells [55,56,57]. There are also data implicating the SCFA receptor FFAR2 in the ability of butyrate to suppress HDAC expression and hypermethylation of inflammation suppressors in colon carcinogenesis, suggesting that this specific receptor could be involved in epigenetic control of tumor suppression [58]. However, in vivo studies in mice revealed that butyrate suppressed colon carcinogenesis through the modulation of immune responses, without affecting HDAC and downstream signaling pathways [86]. In fact, butyrate increased Treg expression acting on GPR109A receptors, reducing proinflammatory cytokine production and preventing cell proliferation and cell survival [50]. 

Valproic acid (2-propylpentanoic acid), a branched SCFA well-known as an anticonvulsant drug, significantly inhibited proliferation of breast cancer cells, in vitro and in pre-clinical studies, by modulating multiple signaling pathways [87]. Valproic acid reduced the cell viability of MCF-7 cells through arrest of the cell cycle, induction of cyclin-dependent kinase inhibitor p21, apoptosis (by upregulation) of Bak, and decreased telomerase activity [88]. This SCFA is clinically available as an HDAC inhibitor, producing cytotoxicity in BC stem cells via apoptosis [89]. These results support targeting epigenetic regulation of histones as a promising treatment of breast cancer.

### 3.2. Modulation of Redox Signaling by SCFAs and Its Impact on Breast Cancer Development

Studies on human colon carcinoma cells confirmed that butyrate inhibited HDAC, leading to the transcriptional upregulation of detoxifying enzymes, such as gluthation-S-transferases (GSTs), and contributing to primary cancer prevention [90]. Butyrate also caused a dose-dependent reduction in inducible nitric oxide synthase transcriptional activity, which decreased NO production via a mechanism independent of HDAC [91]. In colorectal cancer cells, butyrate treatment, at physiological concentrations, induced cell cycle arrest, associated with mitochondria-mediated apoptosis and accompanied by increased mitochondrial superoxide/ROS production [92]. It has also been demonstrated that butyrate modulates both apoptosis and proliferation via miR-22 expression in hepatic cells. It does so by altering the mitochondrial membrane potential, downregulating the HDAC SIRT-1, and increasing superoxide production [93]. Notably, the expression of miR-22 is downregulated in different cancer lines, and has been shown to function as a tumor suppressor in different cancers, including BC, by inhibiting cell proliferation, migration, and invasion [94,95]. Interestingly, targeting miRNA expression using natural compounds has been considered as an alternative to potentiate the efficacy of conventional forms of therapies by increasing their efficacy against cancer progression [96,97]. The detection of additional targets in BC involves the study of HDAC inhibitors to be used in combinatorial therapies. In Table 3, we summarized selected cell culture studies, showing the efficacy of specific SCFAs on BC development. 

Sodium butyrate and sodium propionate inhibited MCF-7 cell proliferation with an elevated level of ROS generation, increased caspase activity, and reduced mitochondrial membrane potential [98]. These effects were dose-dependent. Low and medium levels of both SCFAs induced differentiation and cell-cycle arrest and blockage in G1 growth phase, while higher doses led to massive apoptosis.

Recent data revealed that sodium butyrate exhibited anticancer and genotoxic effects in BC, as demonstrated by the upregulation of antioxidant enzymes in MCF-7 cells [99]. These beneficial effects depended on dose and time of treatment and will require further molecular characterization. Studies in MCF-7 and another BC cell line, MDA-MB-468, established that sodium butyrate induced a dose- and time-dependent cell toxicity related to the cell cycle arrest and induction of apoptosis, with no significant cell toxicity detected in normal breast cells [78]. 

Treatment with either of three SCFAs (acetate, propionate or butyrate) decreased expression and function of intestinal P-gp in rats, possibly via inhibitionof HDAC/NF-κB pathways, and increased intestinal breast cancer resistance protein (BCRP) expression and function, partly via PPARγ activation, contributing to the maintenance of the intestinal barrier [100].

**Table 3 pathogens-12-00486-t003:** Impact of SCFAs in breast cancer under in vitro culture conditions.

Cell Type	SCFA	Treatment	Effect	Major Findings	References
MCF-7 MDA-MB-468	Butyric	0.5–20 mM NaBu, 48–72 h	↓proliferation	↓HDAC2 ↑ histone crotonylation	[78]
MC7	Butyric Propionic	0.5–10 mM NaBu, NaP 24–72 h	↓proliferation ↑apoptosis	↑differentiation Blockage in G1	[98]
MC7	Butyric	1–5 mM NaBu 24–48 h	↓viability	↑GSH ↓SOD ↓NO, H_2_O_2_	[99]
MC7	Butyric	2.5–20 mM NaBu 48–72 h	↓viability ↑apoptosis	↑ROS ↑caspases ↓Δψm	[101]
MDA-MB-231	Hexanoic	0.9–6.5 mM Hexanoic, 48 h	↓proliferation ↑apoptosis	↓CDK2, CDK4 ↑P21	[81]

Abbreviations: GSH: glutathione; CDK2: cyclin-dependent kinase 2; CDK4: cyclin-dependent kinase4; GSH: glutathione; HDAC2: histone deacetylase 2; NaBu: sodium butyrate; NaP: sodium propionate; NO: nitric oxide; P21: cyclin-dependent kinase inhibitor; ROS: reactive oxigen species; SOD: superoxide dismutase; Δψm: mitocondrial membrane potential. Upward arrow (↑) indicates an increase and downward arrow (↓) a decrease in respective measured outcomes.

Preclinical studies supported the anticancer effects of diet-derived SCFAs [74]. Previous studies with a stable commercial butyric acid derivative, monobut-3, showed decreased human tumor take and tumor growth in mice [102]. Butyrate, delivered as tributyrin, significantly inhibited mammary tumor development in a nitrosomethylurea-induced rat model fed a high-polyunsaturated fat diet to promote mammary cancer [103]. Notably, both studies postulated a protective role for butyrate through estrogen-independent pathways. 

Deletion of Gpr109a, a receptor activated by butyrate, increased tumor incidence and activated early onset of mammary tumorigenesis in MMTV-Neu mouse model of spontaneous breast cancer, supporting the hypothesis that GPR109A was a tumor suppressor in mammary glands [104]. In this work, the authors also determined that GPR109A activation in human breast cancer cells inhibited genes involved in cell survival and antiapoptotic signaling. 

Sodium propionate suppressed tumor growth in mice bearing breast cancer cell xenografts by inhibiting JAK2-STAT3 activation, causing cell cycle arrest and increasing ROS generation, leading to p38 activation and apoptosis in breast cancer cells [105]. A synergistic antitumor effect of valproic acid, in combination with capecitabine, was observed in breast cancer xenograft models in mice after inducing thymidine phosphorylase expression. That could be helpful for the treatment of metastatic breast cancer [106]. 

Nutritional intervention, consisting of broccoli sprouts and green tea polyphenols, in Her2/neu transgenic mice that spontaneously developed estrogen receptor-negative mammary tumors decreased tumor volume and increased tumor latency, correlated with changes in gut microbiota and SCFA composition [107]. Of note, this study showed the relevance of the time of consumption of these compounds during the subjects’ lifespan for the composition of gut microbiota and SCFA profiles, as well as in breast cancer prevention.

Treatment with the SCFAs pentanoate and butyrate increased the tumor-suppressive effects of immunotherapy in murine models by enhancing the efficacy of cytotoxic T lymphocytes and chimeric antigen receptor T cells through metabolic and epigenetic reprogramming [108]. These results supported the use of gut microbial metabolite supplementation for future cancer therapy, although clinical implementation will require careful additional investigation [109]. Notably, the intestinal flora in patients with premenopausal breast cancer showed significantly reduced SCFA–producing bacteria levels, as well as reduced levels of key SCFA-producing enzymes, supporting a relevant role for these metabolites [110].

Even though studies evaluating the relationship between BC and gut microbiota are still limited, several clinical trials are underway at the time of this writing, and could provide valuable information regarding risks, responses to treatment, and recurrence [111,112].

### 3.3. Specific SCFAs Potentiate Cancer Therapies via the Keap1-NRF2 Pathway

As previously mentioned, specific SCFAs are modulators of Keap1-NRF2 redox signaling. These work mostly by epigenetic regulation and/or NRF2 nuclear translocation [65]. Notably, in rat intestinal epithelial cells, butyrate treatment increased antioxidant activities via NRF2, but also decreased mRNA and protein levels of the tumor suppressor p53, supporting crosstalk between p53 and NRF2 in the regulation of cellular redox homeostasis [113,114]. 

A link between butyrate regulation of Keap1-NRF2 signaling and anticancer effects was reported in the colorectal cancer cell model HCT116 [115]. Treatment with this SCFA inhibited cancer cell growth by driving metabolic rewiring and epigenetic reprogramming, involving a decrease in NRF2, an increase in the NRF2 negative regulator Keap1, and inhibition of NRF2-target genes, such as as NQO1. 

The antiproliferative properties of the HDAC inhibitor valproic acid have also been linked to NRF2 signaling. This SCFA enhanced the ROS-mediated killing of acute myeloid leukaemia (AML) cells due to low doses of the anticancer BaP treatment. This was based on bezafibrate and medroxyprogesterone acetate, and worked by disabling the NRF2-targeted antioxidant responses [116]. This combined treatment significantly reduced NRF2 protein levels and suppressed its nuclear translocation. Combination treatment with valproic acid and the anticancer agent TRAIL (tumor necrosis factor-related apoptosis-inducing ligand) also synergistically induced apoptotic cell death in TRAIL-resistant papillary thyroid cancer (PTC) cells through caspase activation, as well as cell death (via suppression of Bcl-xL by downregulating NRF2-signaling) [117]. 

It has also been demonstrated that valproic acid sensitizes more hepatocellular carcinoma cells to proton irradiation by increasing proton-induced DNA damage and augmenting proton-induced apoptosis [118]. This SCFA further increased proton-induced production of intracellular ROS and suppressed expression of NRF2. Therefore, NRF2 is a target of valproic radiosensitization, with interesting prospects. 

Collectively, these results supported the idea that targeting NRF2 pathways using SCFAs could impact clinical outcomes of different cancer therapies, particularly when combined with hormonal or traditional chemotherapy agents. 

## 4. New Targets for Triple-Negative Breast Cancer

Triple-negative breast cancer (TNBC) is a subtype of BC that does not express estrogen receptor (ER), progesterone receptor (PR), or human epidermal growth factor receptor 2 (HER-2), and, therefore, is not sensitive to endocrine therapy or HER2 treatment [119]. Development of new treatment strategies is urgently needed due to the invasiveness and high metastatic potential of TNBC, as well as its poor prognosis.

Epigenetic processes control both the initiation and progression of TNBC and, as the expression of HDACs is frequently upregulated, the development of new therapies could involve HDAC inhibitors in combination therapy, e.g., using chemotherapy or other inhibitors [120]. In fact, treatment with specific inhibitors, in combination with kinase inhibitors, autophagy inhibitors, ionizing radiation, or other HDAC inhibitors, is currently being evaluated [121]. 

Some treatments with two HDAC inhibitors have involved the SCFA butyrate. Treatments with suberoylanilidehydroxamic acid and sodium butyrate have shown promising therapeutic outcomes against TNBC, especially in combination with other anticancer agents. A study treating MDA-MB-231 and BT-549 cell lines showed that these inhibitors suppressed cell proliferation, arrested cell cycles at G0/G1 phase, and promoted mitochondrial-related apoptosis by decreasing the phosphorylation and levels of mutant p53 (mtp53) (the most frequently mutated gene in BC) [122]. 

Interestingly, the HDAC inhibitors sodium butyrate and Trichostatin A reciprocally modulated energy metabolism in cancer cells, including TPN cells [123]. Both molecules increased cell oxygen consumption, showing that epigenetic changes associated with acetylation of proteins affected energy metabolism in all cancer cell lines. This further highlighted the relevant role of mitochondria in metastasis. 

## 5. Conclusions and Future Perspectives

Based on emerging data, SCFAs, obtained from the diet or produced by microbiota metabolism, constitute an attractive tool for early prevention and treatment of breast cancer (Figure 1). These beneficial effects are dose- and time-dependent, as established in in vitro studies. Therefore, the use of specific SCFAs as nutritional and therapeutic agents in breast cancer prevention and treatment merits further investigation. 

Numerous studies have supported targeting NRF2 signaling using SCFAs to potentially impact the clinical outcomes of different breast cancer therapies. Notably, SCFA activity mostly depends on specific receptors and transporters [68,69,124]. Therefore, these natural regulators of the Keap1-NRF2 pathway could potentially avoid undesirable effects of widespread activation of NRF2 in healthy, versus cancerous, tissues [21]. Further in vitro and in vivo studies will be necessary to characterize the molecular mechanisms underlying the modulation of NRF2-Keap1 signaling in BC cells by specific SCFAs. 

Notably, SCFAs establish biological crosstalk between metabolomics and epigenomics through dietary habits, linking the microbiota with the maintenance of host redox homeostasis. Nowadays, the ability to reprogram the cancer epigenome is one of the most promising target therapies in the treatment and reversibility of drug resistance [120]. This led to the emerging area of pharmaco(epi)genomics, focused on the clinical implications of epigenetic changes during tumorigenesis and on the identification of new targets for therapies based on epidrugs. Considering the promising efficacy of butyrate in combination therapy with other HDAC inhibitors for the treatment of TNBC, a more accurate understanding of this resistant type of BC and of butyrate’s mechanisms of action could be suggested for the development of new therapeutic strategies.

A final point to make is the importance of performing in vitro studies, with modulators of redox homeostasis, in cells cultured under physiologically relevant O_2_ levels—as have recently been reviewed [125,126]. The majority of studies have cultured cells under hyperoxia (21%, 20.9 kPa O_2_), which is never encountered in vivo, and is known to upregulate NRF2-targeted antioxidant defences. This is a relevant aspect to be considered in modeling preclinical breast cancer with the aim of improving clinical translation; tumors in vivo are exposed to a hypoxic or low-oxygen (1–2% O_2_) environment [127]. Notably, among differentially-expressed genes at different O_2_ levels, those involved in cancer biology are the most affected, highlighting the importance of performing cancer cell research under physiologically relevant O_2_ levels [128]. Recently, it was demonstrated that tumors collected, processed, and propagated at physiological O_2_ levels showed differences in key signaling networks, including Keap1-NRF2, and sensitivity to targeted therapies, compared to those collected, processed, and propagated in ambient air [129]. Therefore, evaluating cancer cells under physioxia will be necessary to recreate the effects of natural bioactive molecules, including SCFAs, under physiopathologic conditions encountered in vivo.

## Figures and Tables

**Figure 1 pathogens-12-00486-f001:**
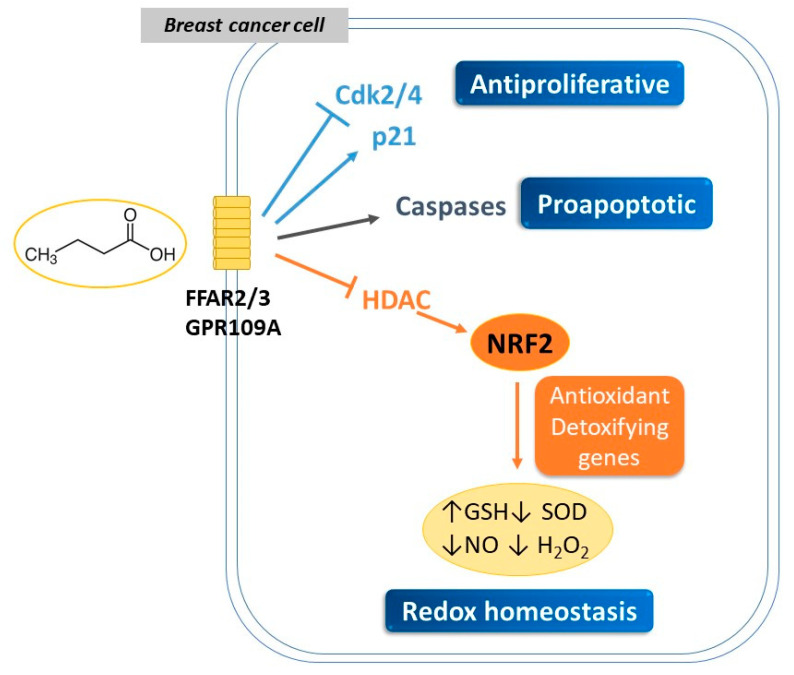
The demonstrated beneficial effects of the SCFA butyrate, produced by microbiota and obtained from the diet, in breast cancer cells in vitro, highlighting redox homeostasis regulation. Cdk2: cyclin-dependent kinase 2; Cdk4: cyclin-dependent kinase; 2FFAR2: free fatty acid receptor 2; FFAR3: free fatty acid receptor 3; GR109A: G-protein-coupled receptor; GSH: glutathione; HDAC: histone deacetylase; HO1: heme oxygenase 1; NO: nitric oxide; NQ1: NAD(P)H: quinone oxidoreductase-1; Nrf2: nuclear erythroid 2-related factor 2; P21: cyclin-dependent kinase inhibitor P21; H_2_O_2_: oxygen peroxyde; SOD: superoxide dismutase.

**Table 2 pathogens-12-00486-t002:** Antiproliferative properties of SCFAs under in vitro culture conditions.

Cell Line	SCFA	Treatment	Major Findings	References
HCT116	Butyric	0.5–5 mM NaB 48 h	↓HDAC2 ↑ histone crotonylation	[78]
HSC-2	Butyric	10 mM Butyrate 24 h	↓ICAM-1 /Nrf2 independent ↓p65 nuclear translocation	[79]
HT29	Butyric, valeric	10% NaB, valeric, 48 h	↓ HDAC2	[80]
HCT-116	Hexanoic	0.9–6.5 mM Hexanoic, 48 h	↓CDK2, CDK4 ↑P21	[81]

Abbreviations: CDK2: cyclin-dependent kinase2; CDK4: cyclin-dependent kinase4; HDAC2: histone deacetylase 2; ICAM-1: intercellular adhesion molecule 1; Nrf2: nuclear erythroid 2-related factor 2; P21: cyclin-dependent kinase inhibitor; p65: transcription factor. Upward arrow (↑) indicates an increase and downward arrow (↓) a decrease in respective measured outcomes.

## Data Availability

Not applicable.

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
