# Peer review of "Control of Redox Homeostasis by Short-Chain Fatty Acids: Implications for the Prevention and Treatment of Breast Cancer"

_pathogens, 2023, doi:10.3390/pathogens12030486_

Round 1
Reviewer 1 Report
In this manuscript, the authors reviewed the role of short-chain fatty acids on redox homeostasis in breast cancer cells. Here are a few additional minor suggestions.
1 Previous studies showed that some dietary derived short chain fatty acids (butyrate, propionate and acetate) exhibit HDAC inhibitory activity and also enhance Histone 3 K9 acetylation. It would be better to include the references of H3K9Ac.
2 It would be better to add more references about patient studies and animal models. For example, butyrate inhibits breast tumor growth in vivo.
Author Response
Dear reviewer:
Thank you for your comments.
Following your suggestions we have addressed the specific changes as follows:
- We have mentioned that SCFAs enhance H3K9 acetylation and included references.
- We have included data and references of studies with animal models supporting that specific SCFAs inhibit breast tumor growth in vivo. We also discuss that studies evaluating the relationship between BC and gut microbiota are quite limited, but related clinical trials are underway that will provide valuable information about the role of microbiota and SCFAs in the prevention and treatment of breast cancer.
- The manuscript has been checked by a native English-speaking colleague.
Reviewer 2 Report
It’s a well-written review on the antioxidant properties of SCFAs and their impact on cancer development and treatment with special emphasis on breast cancer. I don’t have any concerns.
Author Response
Dear reviewer:
Thank you very much for your comments.
Following your suggestions, the manuscript has been checked by a native English-speaking colleague.